# Thompson Sampling with Approximate Inference

**My Phan**
College of Information and Computer Science
University of Massachusetts
Amherst, MA
myphan@cs.umass.edu

**Yasin Abbasi-Yadkori**
VinAI
Hanoi, Vietnam
yasin.abbasi@gmail.com

**Justin Domke**
College of Information and Computer Science
University of Massachusetts
Amherst, MA
domke@cs.umass.edu

## Abstract

We study the effects of approximate inference on the performance of Thompson sampling in the $k$-armed bandit problems. Thompson sampling is a successful algorithm for online decision-making but requires posterior inference, which often must be approximated in practice. We show that even small constant inference error (in $\alpha$-divergence) can lead to poor performance (linear regret) due to under-exploration (for $\alpha < 1$) or over-exploration (for $\alpha > 0$) by the approximation. While for $\alpha > 0$ this is unavoidable, for $\alpha \leq 0$ the regret can be improved by adding a small amount of forced exploration even when the inference error is a large constant.

## 1 Introduction

The stochastic $k$-armed bandit problem is a sequential decision making problem where at each time-step $t$, a learning agent chooses an action (arm) among $k$ possible actions and observes a random reward. Thompson sampling (Russo et al., 2018) is a popular approach in bandit problems based on sampling from a posterior in each round. It has been shown to have good performance both in term of frequentist regret and Bayesian regret for the $k$-armed bandit problem under certain conditions.

This paper investigates Thompson sampling when only an *approximate* posterior is available. This is motivated by the fact that in complex models, approximate inference methods such as Markov Chain Monte Carlo or Variational Inference must be used. Along this line, Lu & Van Roy (2017) propose a novel inference method – Ensemble sampling – and analyze its regret for linear contextual bandits. To the best of our knowledge this is the most closely related theoretical analysis of Thompson sampling with approximate inference.

This paper analyzes the regret of Thompson sampling with approximate inference. Rather than considering a particular inference algorithm, we parameterize the error using the $\alpha$-divergence, a typical measure of inference accuracy. Our contributions are as follows:

- **Even small inference errors can lead to linear regret with naive Thompson sampling.** Given any error threshold $\epsilon > 0$ and any $\alpha$ we show that approximate posteriors with error at most $\epsilon$ in $\alpha$-divergence at all times can result in linear regret (both frequentist and Bayesian). For $\alpha > 0$ and for any reasonable prior, we show linear regret due to over-exploration by the approximation (Theorem 1, Corrolary 1). For $\alpha < 1$ and for priors satisfying certain

conditions, we show linear regret due to under-exploration by the approximation, which prevents the posterior from concentrating (Theorem 2, Corrolary 2).

- **Forced exploration can restore sub-linear regret.** For $\alpha \leq 0$ we show that adding forced exploration to Thompson sampling can make the posterior concentrate and restore sub-linear regret (Theorem 3) even when the error threshold is a very large constant. We illustrate this effect by showing that the performances of Ensemble sampling (Lu & Van Roy, 2017) and mean-field Variation Inference (Blei et al., 2017) can be improved in this way either theoretically (Section 5.1) or in simulations (Section 6).

## 2 Background and Notations.

### 2.1 The $k$-armed Bandit Problem.

We consider the $k$-armed bandit problem parameterized by the mean reward vector $m = (m_1, ..., m_k) \in \mathcal{R}^k$, where $m_i^*$ denotes the mean reward of arm (action) $i$. At each round $t$, the learner chooses an action $A_t$ and observes the outcome $Y_t$ which, conditioned on $A_t$, is independent of the history up to and not including time $t$, $H_{t-1} = (A_1, Y_1, ..., A_{t-1}, Y_{t-1})$. For a time horizon $T$, the goal of the algorithm $\pi$ is to maximize the expected cumulative reward up to time $T$.

Let $\Omega \subseteq \mathcal{R}^k$ be the domain of the mean and $\Omega_i \subseteq \Omega$ denote the region where the $i$th arm has the largest mean. Let the function $A^* : \Omega \to \{a_1, ..., a_k\}$ denoting the best action be defined as: $A^*(m) = i$ if $m \in \Omega_i$.

In the frequentist setting we assume that there exists a true mean $m^*$ which is fixed and unknown to the learner. Therefore, a policy $\pi^*$ that always chooses $A^*(m^*)$ will get the highest reward. The performance of policy $\pi$ is measured by its expected regret compared to an optimal policy $\pi^*$, which is defined as:

$$\text{Regret}(T, \pi, m^*) = T m_{A^*(m^*)}^* - \mathbb{E} \sum_{t=1}^{T} m_{A_t}^* . \tag{1}$$

On the other hand, in the Bayesian setting, an agent expresses her beliefs about the mean vector in terms of a prior $\Pi_0$, and therefore, the mean is treated as a random variable $M = (M_1, ..., M_k)$ distributed according to the prior $\Pi_0$. The Bayesian regret is the expectation of the regret under the prior of parameter $M$:

$$\text{BayesRegret}(T, \pi) = \mathbb{E}_{\Pi_0} \text{Regret}(T, \pi, M) . \tag{2}$$

### 2.2 Thompson Sampling with Approximate Inference

In the frequentist setting, in order to perform Thompson sampling we define a prior which is only used in the algorithm. On the other hand, in the Bayesian setting the prior is given.

Let $\Pi_t$ be the posterior distribution of $M|H_{t-1}$ with density function $\pi_t(m)$. Thompson sampling obtains a sample $\widehat{m}$ from $\Pi_t$ and then selects arm $A_t$ as follow: $A_t = i$ if $\widehat{m} \in \Omega_i$. In each round, we assume an approximate sampling method is available that generates sample from an approximate distribution $Q_t$. We use $q_t$ to denote the density function of $Q_t$.

Popular approximate sampling methods include Markov Chain Monte Carlo (MCMC) (Andrieu et al., 2003), Sequential Monte Carlo (Doucet & Johansen, 2009) and Variational Inference (VI) (Blei et al., 2017). There are packages that conveniently implement VI and MCMC methods, such as Stan (Carpenter et al., 2017), Edward (Tran et al., 2016), PyMC (Salvatier et al., 2016) and infer.NET (Minka et al., 2018).

To provide a general analysis of approximate sampling methods, we will use the $\alpha$-divergence (Section 2.3) to quantify the distance between the posterior $\Pi_t$ and the approximation $Q_t$.

## 2.3 The Alpha Divergence

The $\alpha$-divergence between two distributions $P$ and $Q$ with density functions $p(x)$ and $q(x)$ is defined as:

$$D_\alpha(P, Q) = \frac{1 - \int p(x)^\alpha q(x)^{1-\alpha} dx}{\alpha(1 - \alpha)}. \tag{3}$$

$\alpha$-divergence generalizes many divergences, including $KL(Q, P)$ ($\alpha \to 0$), $KL(P, Q)$ ($\alpha \to 1$), Hellinger distance ($\alpha = 0.5$) and $\chi^2$ divergence ($\alpha = 2$) and is a common way to measure errors in inference methods. MCMC errors are measured by the Total Variation distance, which can be upper bounded by the KL divergence using Pinsker's inequality ($\alpha = 0$ or $\alpha = 1$). Variational Inference tries to minimize the reverse KL divergence (information projection) between the target distribution and the approximation ($\alpha = 0$). Ensemble sampling (Lu & Van Roy, 2017) provides error guarantees using reverse KL divergence ($\alpha = 0$). Expectation Propagation tries to minimize the KL divergence ($\alpha = 1$) and $\chi^2$ Variational Inference tries to minimize the $\chi^2$ divergence ($\alpha = 2$).

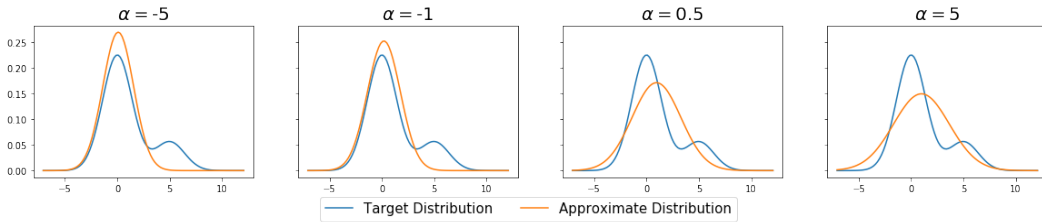

Figure 1: The Gaussian $Q$ which minimizes $D_\alpha(P, Q)$ for different values of $\alpha$ where the target distribution $P$ is a mixture of two Gaussians. Based on Figure 1 from (Minka, 2005)

When $\alpha$ is small, the approximation fits the posterior's dominant mode. When $\alpha$ is large, the approximation covers the posterior's entire support (Minka, 2005) as illustrated in Figure 1. Therefore changing $\alpha$ will affect the exploration-exploitation trade-off in bandit problems.

## 2.4 Problem Statement.

**Problem Statement.** For the $k$-armed bandit problem, given $\alpha$ and $\epsilon > 0$, if at all time-steps $t$ we sample from an approximate distribution $Q_t$ such that $D_\alpha(\Pi_t, Q_t) < \epsilon$, will the regret be sub-linear in $t$?

# 3 Motivating Example

In this section we present a simple example to show the effects of inference errors on the frequentist regret.

**Example.** Consider a 2-armed bandit problem where the reward distributions are $\text{Norm}(0.6, 0.2^2)$ and $\text{Norm}(0.5, 0.2^2)$ for arm 1 and 2 respectively. The prior $\Pi_0$ is $\text{Norm}\left(\mu_0^T, 0.5^2 I\right)$ where $\mu_0 = [0.1, 0.9]$ is the vector of prior means of arm 1 and 2 respectively, and $I$ denotes the identity matrix.

Let $\Pi_t = \text{Norm}(\mu_t, \Sigma_t)$ be the posterior at time $t$. Approximations $Q_t$ and $Z_t$ are calculated such that $\text{KL}(\Pi_t, Q_t) = 2$ and $\text{KL}(Z_t, \Pi_t) = 1.5$ by multiplying the covariance $\Sigma_t$ by a constant: $Q_t = \text{Norm}(\mu_t, 4.5^2 \Sigma_t)$ and $Z_t = \text{Norm}(\mu_t, 0.3^2 \Sigma_t)$. The KL divergence between two Gaussian distributions is provided in Appendix F.

We perform the following simulations 1000 times and plot the mean cumulative regret up to time $T = 100$ in Figure 2b using three different policies:

1. (**Exact Thompson Sampling**) At each time-step $t$, sample from the true posterior $\Pi_t$.
2. (**Approximation $Q_t$**) At each time-step $t$, compute $Q_t$ from $\Pi_t$ and sample from $Q_t$.
3. (**Approximation $Z_t$**) At each time-step $t$, compute $Z_t$ from $\Pi_t$ and sample from $Z_t$.

The regrets of sampling from the approximations $Q_t$ and $Z_t$ are in both cases larger than that of exact Thompson sampling. Intuitively, the regret of $Q_t$ is larger because $Q_t$ explores more than the true

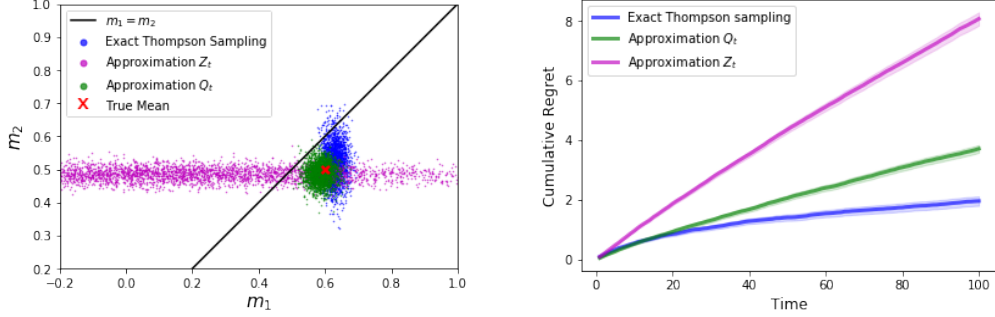

(a) Over-dispersed (approximation $Q_t$) and under-dispersed sampling (approximation $Z_t$) yield different posteriors after $T = 100$ time-steps. $m_1$ and $m_2$ are the means of arms 1 and 2. $Q_t$ picks arm 2 more often than exact Thompson sampling and $Z_t$ mostly picks arm 2. The posteriors of exact Thompson sampling and $Q_t$ concentrate mostly in the region where $m_1 > m_2$ while $Z_t$'s spans both regions.

(b) The regret of sampling from the approximations $Q_t$ and $Z_t$ are both larger than that of exact Thompson sampling from the true posterior $\Pi_t$. Shaded regions show 95% confidence intervals.

Figure 2: Approximation $Q_t$ (with high variance) and approximation $Z_t$ (with small variance) are defined in Section 3 where $D_1(\Pi_t, Q_t) = 2$ and $D_0(\Pi_t, Z_t) = 1.5$. Arm 1 is the true best arm.

posterior (Figure 2a). In Section 4 we show that when $\alpha > 0$ the approximation can incur this type of error, leading to linear regret. On the other hand, the regret of $Z_t$ is larger because $Z_t$ explores less than the exact Thompson sampling algorithm and therefore commits to the sub-optimal arm (Figure 2a). In Section 5 we show that when $\alpha < 1$ the approximation can change the posterior concentration rate, leading to linear regret. We also show that adding a uniform sampling step can help the posterior to concentrate when $\alpha \leq 0$, and make the regret sub-linear.

## 4 Regret Analysis When $\alpha > 0$

In this section we analyze the regret when $\alpha > 0$. Our result shows that the approximate method might pick the sub-optimal arm with constant probability in every time-step, leading to linear regret.

**Theorem 1** (Frequentist Regret). *Let $\alpha > 0$, the number of arms be $k = 2$ and $m_1^* > m_2^*$. Let $\Pi_0$ be a prior where $\mathbb{P}_{\Pi_0}(M_2 > M_1) > 0$. For any error threshold $\epsilon > 0$, there is a deterministic mapping $f(\Pi)$ such that for all $t \geq 0$:*

*1. Sampling from $Q_t = f(\Pi_t)$ chooses arm 2 with a constant probability.*
*2. $D_\alpha(\Pi_t, Q_t) < \epsilon$.*

*Therefore sampling from $Q_t$ for $T/10$ time-steps and using any policy for the remaining time-steps will cause linear frequentist regret.*

Typically, approximate inference methods minimize divergences. Broadly speaking, this theorem shows that making a divergence a small constant, alone, is not enough to guarantee sub-linear regret. We do not mean to imply that low regret is *impossible* but simply that making an $\alpha$-divergence a small constant alone is not sufficient.

At every time-step, the mapping $f$ constructs the approximation $Q_t$ from the posterior $\Pi_t$ by moving probability mass from the region $\Omega_1$ where $m_1 > m_2$ to the region $\Omega_2$ where $m_2 > m_1$. Then $Q_t$ will choose arm 2 with a constant probability at every time-step. The constant average regret per time-step is discussed in Appendix A.4.

Therefore, if we sample from $Q_t = f(\Pi_t)$ for $0.1T$ time steps and use any policy in the remaining $0.9T$ time steps, we will still incur linear regret from the $0.1T$ time-steps. On the other hand, when $\alpha \leq 0$, we show in Section 5.1 that sampling an arm uniformly at random for $\log T$ time-steps and sampling from an approximate distribution that satisfies the divergence constraint for $T - \log T$ time-steps will result in sub-linear regret.

Agrawal & Goyal (2013) show that the frequentist regret of exact Thompson sampling is $O(\sqrt{T})$ with Gaussian or Beta priors and bounded rewards. Theorem 1 implies that when the assumptions in (Agrawal & Goyal, 2013) are satisfied but there is a small constant inference error at every time-step, the regret is no longer guaranteed to be sub-linear.

If the assumption $m_1^* > m_2^*$ in Theorem 1 is satisfied with a non-zero probability ($\mathbb{P}_{\Pi_0}(M_1 > M_2) > 0$), the Bayesian regret will also be linear:

**Corollary 1** (Bayesian Regret). *Let $\alpha > 0$ and the number of arms be $k = 2$. Let $\Pi_0$ be a prior where $\mathbb{P}_{\Pi_0}(M_1 > M_2) > 0$ and $\mathbb{P}_{\Pi_0}(M_2 > M_1) > 0$. Then for any error threshold $\epsilon > 0$, there is a deterministic mapping $f(\Pi)$ such that for all $t \geq 0$ the two statements in Theorem 1 hold.*

*Therefore sampling from $Q_t$ for $T/10$ time-steps and using any policy for the remaining time-steps will cause linear Bayesian regret.*

Russo & Roy (2016) prove that the Bayesian regret of Thompson sampling for $k$-armed bandits with sub-Gaussian rewards is $O(\sqrt{T})$. Corollary 1 implies that even when the assumptions in Russo & Roy (2016) are satisfied, under certain conditions and with approximation errors, the regret is no longer guaranteed to be sub-linear.

## 5  Regret Analysis When $\alpha < 1$

In this section we analyze the regret when $\alpha < 1$. Our result shows that for any error threshold, if the posterior $\Pi_t$ places too much probability mass on the wrong arm then the approximation $Q_t$ is allowed to avoid the optimal arm. If the sub-optimal arms do not provide information about the arms' ranking, the posterior $\Pi_{t+1}$ does not concentrate. Therefore $Q_{t+1}$ is also allowed to be close in $\alpha$-divergence while avoiding the optimal arm, leading to linear regret in the long term.

**Theorem 2** (Frequentist Regret). *Let $\alpha < 1$, the number of arms be $k = 2$ and $m_1^* > m_2^*$. Let $\Pi_0$ be a prior where $M_2$ and $M_1 - M_2$ are independent. There is a deterministic mapping $f(\Pi)$ such that for all $t \geq 0$:*

1. *Sampling from $Q_t = f(\Pi_t)$ chooses arm 2 with probability 1.*
2. *For any $\epsilon > 0$, there exists $0 < z \leq 1$ such that if $\mathbb{P}_{\Pi_0}(M_2 > M_1) = z$ and arm 2 is chosen at all times before $t$ then $D_\alpha(\Pi_t, Q_t) < \epsilon$.*
   *For any $0 < z \leq 1$, there exists $\epsilon > 0$ such that if $\mathbb{P}_{\Pi_0}(M_2 > M_1) = z$ and arm 2 is chosen at all times before $t$ then $D_\alpha(\Pi_t, Q_t) < \epsilon$.*

*Therefore sampling from $Q_t$ at all time-steps results in linear frequentist regret.*

We discuss why the above results are not immediately obvious. When $\alpha \to 0$, the $\alpha$-divergence becomes $\mathrm{KL}(Q_t, \Pi_t)$. We might believe that the regret should be sub-linear in this case because the posterior $\Pi_t$ becomes more concentrated, and so the total variation between $Q_t$ and $\Pi_t$ must decrease. For example, Ordentlich & Weinberger (2004) show the distribution-dependent Pinsker's inequality between $\mathrm{KL}(Q, P)$ and the total variation $\mathrm{TV}(P, Q)$ for discrete distributions $P$ and $Q$ as follows:

$$\mathrm{KL}(Q, P) \geq \phi(P) \cdot \mathrm{TV}(P, Q)^2 \ . \tag{4}$$

Here, $\phi(P)$ is a quantity that will increase to infinity if $P$ becomes more concentrated. However, the algorithm in Theorem 2 constructs an approximation distribution that never picks the optimal arm, so the posterior $\Pi_t$ can not concentrate and the regret is linear. The error threshold $\epsilon$ causing linear frequentist regret is correlated with the probability mass the prior places on the true best arm (Appendix B.4).

With some assumptions on the rewards, Gopalan et al. (2014) show that the problem-dependent frequentist regret is $O(\log T)$ for finitely-supported, correlated priors with $\pi_0(m^*) > 0$. Liu & Li (2016) study the prior-dependent frequentist regret of 2-armed-and-2-models bandits, and show that with some smoothness assumptions on the reward likelihoods, the regret is $O(\sqrt{T/\mathbb{P}_{\Pi_0}(M_2 > M_1)})$ if arm 1 is the better arm. Theorem 2 implies that when the assumptions in (Gopalan et al., 2014) or (Liu & Li, 2016) are satisfied, if $M_2$ and $M_1 - M_2$ are independent and there are approximation errors, the regret is no longer guaranteed to be sub-linear.

If the assumption $m_1^* > m_2^*$ in Theorem 2 is satisfied with a non-zero probability ($\mathbb{P}_{\Pi_0}(M_1 > M_2) > 0$), the Bayesian regret wil also be linear:

**Corollary 2** (Bayesian Regret). *Let $\alpha < 1$ and the number of arms be $k = 2$. Let $\Pi_0$ be a prior where $\mathbb{P}_{\Pi_0}(M_1 > M_2) > 0$ and $M_2$ and $M_1 - M_2$ are independent. There is a deterministic mapping $f(\Pi)$ such that for all $t \geq 0$ the 2 statements in Theorem 2 hold.*

*Therefore sampling from $Q_t$ at all time-steps results in linear Bayesian regret.*

Russo & Roy (2016) prove that the Bayesian regret of Thompson sampling for $k$-armed bandits with sub-Gaussian rewards is $O(\sqrt{T})$. Corollary 2 implies that even when the assumptions in Russo & Roy (2016) are satisfied, under certain conditions and with approximation errors, the regret is no longer guaranteed to be sub-linear.

We note that, unlike the case when $\alpha > 0$, if we use another policy in $o(T)$ time-steps to make the posterior concentrate and sample from $Q_t$ for the remaining time-steps, the regret can be sub-linear. We provide a concrete algorithm in Section 5.1 for the case when $\alpha \leq 0$.

### 5.1 Algorithms with Sub-linear Regret for $\alpha \leq 0$

In the previous section, we see that when $\alpha < 1$, the approximation has linear regret because the posterior does not concentrate. In this section we show that when $\alpha \leq 0$, it is possible to achieve sub-linear regret even when $\epsilon$ is a very large constant by adding a simple exploration step to force the posterior to concentrate (the case of $\alpha > 0$ cannot be improved according to Theorem 1). We first look at the necessary and sufficient condition that will make the posterior concentrate, and then provide an algorithm that satisfies it. Russo (2016) and Qin et al. (2017) both show the following result under different assumptions:

**Lemma 1** (Lemma 14 from Russo (2016)). *Let $m^* \in \mathcal{R}^k$ be the true parameter and let $a^* = A^*(m^*)$ be the true best arm. If for all arms $i$, $\sum_{t=1}^{\infty} P(A_t = i|H_{t-1}) = \infty$, then*

$$\lim_{t \to \infty} P(A^*(M) = a^*|H_{t-1}) = 1 \text{ with probability } 1 . \tag{5}$$

*If there exists arm $i$ such that $\sum_{t=1}^{\infty} P(A_t = i|H_{t-1}) < \infty$, then $\liminf_{t \to \infty} P(A^*(M) = i|H_{t-1}) > 0$ with probability 1.*

Russo (2016) make the following assumptions, which allow correlated priors:

**Assumption 1.** *Let the reward distributions be in the canonical one dimensional exponential family with the density: $p(y|m) = b(y)\exp(mT(y) - A(m))$ where $b, T$ and $A$ are known function and $A(m)$ is assumed to be twice differentiable. The parameter space $\Omega = (\overline{m}, \underline{m})$ is a bounded open hyper-rectangle, the prior density is uniformly bounded with $0 < \inf_{m \in \Omega} \pi_0(m) < \sup_{m \in \Omega} \pi_0(m) < \infty$ and the log-partition function has bounded first derivative with $\sup_{\theta \in [\overline{m}, \underline{m}]} |A'(m)| < \infty$.*

Qin et al. (2017) make the following assumptions:

**Assumption 2.** *Let the prior be an uncorrelated multivariate Gaussian. Let the reward distribution of arm $i$ be $\mathrm{Norm}(m_i, \sigma^2)$ with a common known variance $\sigma^2$ but unknown mean $m_i$.*

Even though we consider the error in sampling from the posterior distribution, the regret is a result of choosing the wrong arm. We define $\overline{\Pi}_t$ as the posterior distribution of the best arm and $\overline{Q}_t$ as the approximation of $\overline{\Pi}_t$ with the density functions

$$\overline{\pi}_t(i) = P(A^* = i|H_{t-1}) \text{ and } \overline{q}_t(i) = P(A_t = i|H_{t-1}).$$

We now define an algorithm where each arm will be chosen infinitely often, satisfying the condition of Lemma 1.

**Theorem 3** (Bayesian and Frequentist Regret). *Consider the case when Assumption 1 or 2 is satisfied. Let $\alpha \leq 0$ and $p_t = o(1)$ be such that $\sum_{t=1}^{\infty} p_t = \infty$. For any number of arms $k$, any prior $\Pi_0$ and any error threshold $\epsilon > 0$, the following algorithm has $o(T)$ frequentist regret: at every time-step $t$,*

- *with probability $1 - p_t$, sample from an approximate posterior $Q_t$ such that $D_\alpha(\overline{\Pi}_t, \overline{Q}_t) < \epsilon$,*
- *with probability $p_t$, sample an arm uniformly at random.*

*Since the Bayesian regret is the expectation of the frequentist regret over the prior, for any prior if the frequentist regret is sub-linear at all points the Bayesian regret will be sub-linear.*

The following lemma shows that the error in choosing the arms is upper bounded by the error in choosing the parameters. Therefore whenever the condition $D_\alpha(\Pi_t, Q_t) < \epsilon$ is satisfied, the condition $D_\alpha(\overline{\Pi}_t, \overline{Q}_t) < \epsilon$ will be satisfied and Theorem 3 is applicable.

**Lemma 2.**
$$D_\alpha(\overline{\Pi}_t, \overline{Q}_t) \leq D_\alpha(\Pi_t, Q_t).$$

We also note that we can achieve sub-linear regret even when $\epsilon$ is a very large constant. We revisit Eq. 4 to provide the intuition: $\mathrm{KL}(Q, P) \geq \phi(P) \cdot \mathrm{TV}(P, Q)^2$. Here, $\phi(P)$ is a quatity that will increase to infinity if $P$ becomes more concentrated. Hence, if $KL(\overline{Q}_t, \overline{\Pi}_t) < \epsilon$ for any constant $\epsilon$ and $\overline{\Pi}_t$ becomes concentrated, the total variation $\mathrm{TV}(\overline{Q}_t, \overline{\Pi}_t)$ will decrease. Therefore, $\overline{Q}_t$ will become concentrated, resulting in sub-linear regret.

**Application.** Lu & Van Roy (2017) propose an approximate sampling method called Ensemble sampling where they maintain a set of $\mathcal{M}$ models to approximate the posterior and analyze its regret for the linear contextual bandits when $\mathcal{M}$ is $\Omega(\log(T))$. For the $k$-armed bandit problem and when $\mathcal{M}$ is $\Theta(\log(T))$, Ensemble sampling satisfies the condition $\mathrm{KL}(\overline{Q}_t, \overline{\Pi}_t) < \epsilon$ in Theorem 3 with high probability. In this case, Lu & Van Roy (2017) show a regret bound that scales linearly with $T$. We discuss in Appendix E how to apply Theorem 3 to get sub-linear regret with Ensemble sampling when $\mathcal{M}$ is $\Theta(\log(T))$.

## 6 Simulations

For each approximation method we repeat the following simulations for 1000 times and plot the mean cumulative regret, using five different policies.

1. (**Exact Thompson sampling**) Use exact posterior sampling to choose an action and update the posterior (for reference).
2. (**Approximation method**) Use the approximation method to choose an action and update the posterior. We use the approximation naively without any modification.
3. (**Forced Exploration**) With a probability (the exploration rate), choose an action uniformly at random and update the posterior. Otherwise, use the approximation method to choose an action and update the posterior. This is the method suggested by Thm. 3.
4. (**Approximate Sample**) Use the approximation method to choose an action. Use exact posterior sampling to update the posterior.
5. (**Approximate Update**) Use exact posterior sampling to choose an action. Use the approximate method to update the posterior.

The last two policies are performed to understand how the approximation affects the posterior (discussed in Section 6.3). We update the posterior using the closed-form formula when both the prior and reward distribution are Gaussian in Appendix G.

### 6.1 Adding Forced Exploration to the Motivating Example

In this section we revisit the example in Section 3. We apply $Q_t$, $Z_t$ and Ensemble sampling with $\mathcal{M} = 2$ models to the bandit problem described in the example. We set the exploration rate at time $t$ to be $1/t$, $T = 100$ and show the results in Figure 3a and discuss them in Section 6.3.

### 6.2 Simulations of Ensemble Sampling and Variational Inference for $50$-armed bandits

Now we add forced exploration to mean-field Variational Inference (VI) and Ensemble Sampling with $\mathcal{M} = 5$ models for a 50-armed bandit instance. We generate the prior and the reward distribution as follows: the prior is $\mathrm{Norm}(\mathbf{0}, \Sigma_0)$. To generate a positive semi-definite matrix $\Sigma_0$, we generate a random matrix $A$ of size $(k, k)$ where entries are uniformly sampled from $[0, 1)$ and set $\Sigma_0 = A^T A / k$. The true mean $m^*$ is sampled from the prior. The reward distribution of arm $i$ is $\mathrm{Norm}(m_i^*, 1)$.

Mean-field VI approximates the posterior by finding an uncorrelated multivariate Gaussian distribution $Q_t$ that minimizes $KL(\Pi_t, Q_t)$. If the posterior is $\Pi_t = \mathrm{Norm}(\mu_t, \Sigma_t)$ then $Q_t$ has the closed-form solution $Q_t = \mathrm{Norm}(\mu_t, \mathrm{Diag}(\Sigma_t^{-1})^{-1})$, which we used to perform the simulations. We set the exploration rate at time $t$ to be $50/t$, $T = 3000$, show the results in Figure 3b and discuss them in Section 6.3.

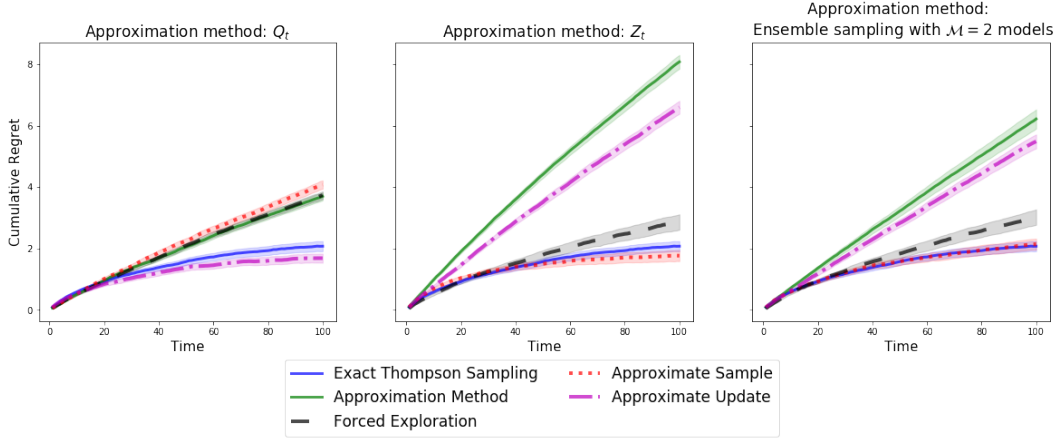

(a) Applying approximations $Q_t$, $Z_t$ and Ensemble Sampling to the motivating example (Section 6.1).

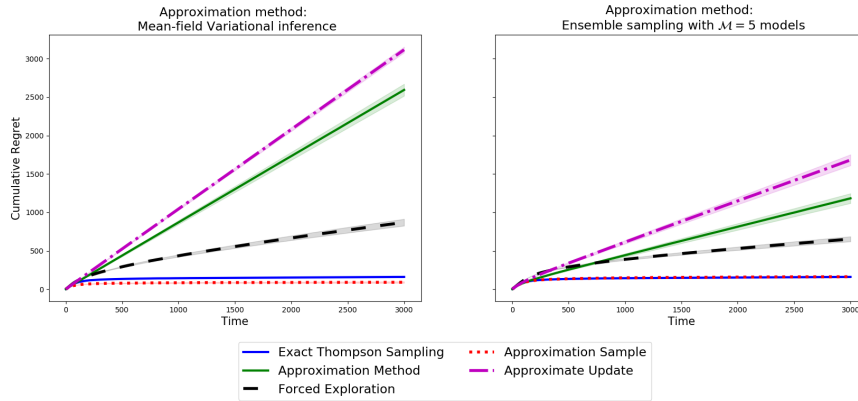

(b) Applying mean-field Variational Inference (VI) and Ensemble sampling on a 50-armed bandit (Section 6.2).

Figure 3: Updating the posterior by exact Thompson sampling or adding forced exploration does not help the over-explored approximation $Q_t$, but lowers the regrets of the under-explored approximations $Z_t$, Ensemble sampling and mean-field VI. Shaded regions show 95% confidence intervals.

## 6.3 Discussion

We observe in Figure 3a that the regret of $Q_t$ calculated from the posterior updated by exact Thompson sampling does not change significantly. Moreover, exact posterior sampling with the posterior updated by $Q_t$ has the same regret as exact Thompson sampling. These two observations imply that $Q_t$ has the same effect on the posterior as exact Thompson sampling. Therefore adding forced exploration is not helpful.

On the other hand, in Figures 3a and 3b the regrets of $Z_t$, Ensemble sampling and mean-field VI calculated from the posterior updated by exact Thompson sampling decrease significantly. Moreover, exact posterior sampling with the posterior updated by the approximations has similar regret to using the approximations. This behaviour is likely because the approximation causes the posterior to concentrate in the wrong region[1]. In combination, these two observations suggest that these methods do not explore enough for the posterior to concentrate. Therefore adding forced exploration is helpful, which is compatible with the result in Theorem 3.

## 7 Related Work

There have been many works on sub-linear Bayesian and frequentist regrets for exact Thompson sampling. We discussed relevant works in detail in Section 4 and Section 5.

Ensemble sampling (Lu & Van Roy, 2017) gives a theoretical analysis of Thompson sampling with one particular approximate inference method. Lu & Van Roy (2017) maintain a set of $\mathcal{M}$ models to approximate the posterior, and analyzed its regret for linear contextual bandits when $\mathcal{M}$ is $\Omega(\log(T))$. For the $k$-armed bandit problem and when $\mathcal{M}$ is $\Theta(\log(T))$, Ensemble sampling satisfies the condition $\mathrm{KL}(\overline{Q}_t, \overline{\Pi}_t) < \epsilon$ in Theorem 3 with high probability. In this case, the regret of Ensemble sampling scales linearly with $T$.

We show in Theorem 2 that when the constraint $\mathrm{KL}(Q_t, \Pi_t) < \epsilon$ is satisfied, which implies by Lemma 2 that $\mathrm{KL}(\overline{Q}_t, \overline{\Pi}_t) < \epsilon$ is satisfied, there can exist approximation algorithms that have linear regret in $T$. This result provides a linear lower bound, which is complementary with the linear regret upper bound of Ensemble Sampling in (Lu & Van Roy, 2017). Moreover, we show in Appendix E that we can apply Theorem 3 to get sub-linear regret with Ensemble sampling with $\Theta(\log(T))$ models.

In reinforcement learning, there is a notion that certain approximations are "stochastically optimistic" and that this has implications for regret (Osband et al., 2016). This is similar in spirit to our analysis in terms of $\alpha$-divergence, in that the characteristics of inference errors are important.

There has been a number of empirical works using approximate methods to perform Thompson sampling. Riquelme et al. (2018) implement variational inference, MCMC, Gaussian processes and other methods on synthetic and real world data sets and measure the regret. Urteaga & Wiggins (2018) derive a variational method for contextual bandits. Kawale et al. (2015) use particle filtering to implement Thompson sampling for matrix factorization.

Finally, if exact inference is not possible, it remains an open question if it is better to use Thompson sampling with approximate inference, or to use a different bandit method that does not require inference with respect to the posterior. For example Kveton et al. (2019) propose an algorithm based on the bootstrap.

## 8 Conclusion

In this paper we analyzed the performance of approximate Thompson sampling when at each time-step $t$, the algorithm obtains a sample from an approximate distribution $Q_t$ such that the $\alpha$-divergence between the true posterior and $Q_t$ remains at most a constant $\epsilon$ at all time-steps.

Our results have the following implications. To achieve a sub-linear regret, we can only use $\alpha > 0$ for $o(T)$ time-steps. Therefore we should use $\alpha \leq 0$ with forced exploration to make the posterior concentrate. This method theoretically guarantees a sub-linear regret even when $\epsilon$ is a large constant.

### Acknowledgments

We thank Huy Le for providing the proof of Lemma 9.

## Footnotes

[1]Note that in the case where there are 2 arms (Figure 3a), exact posterior sampling with the posterior updated by the approximate method has slightly lower regret than naively using the approximate method. This is only because there are only 2 regions, so exact posterior sampling explores more than the approximation in the other region, which happens to be the correct one.

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
