[Supplementary Material]

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

# A Proof of Theorem 1 and Corollary 1

First we will prove Theorem 1. Let $\Omega_i \subseteq \Omega$ denote the region where arm $i$ is the best arm. Let $\Pi_{t,i}$ denote $\Pi_t(\Omega_i)$, the posterior probability that arm $i$ is the best arm. For $r > 1$, We construct the pdf of $Q_t$'s as follows:

$$q_t(m) = \begin{cases} \frac{1}{r}\pi_t(m), & \text{if } m_1 > m_2 \\ \frac{1-\Pi_{t,1}/r}{1-\Pi_{t,1}}\pi_t(m), & \text{otherwise.} \end{cases} \tag{6}$$

We will prove the theorem by the following steps:

- In Lemma 3 we show that $Q_t$'s are valid distributions.

- In Lemma 4 we show that when $\alpha > 0$ the $\alpha$-divergence between $Q_t$ and $\Pi_t$ can be arbitrarily small

- In Lemma 5 we show that sampling from $Q_t$ for $\Theta(T)$ time-steps will generate linear frequentist regret, and lower bound the regret.

Since the regret is linear, in Appendix A.4 we discuss the constant average regret per time-step as a function of $\epsilon$ and $\alpha$. In Appendix A.5 we provide the Bayesian regret proof for Corollary 1.

**Lemma 3.** $q_t(m)$ in Eq. 6 is well-defined and if $\int \pi_t(m)dm = 1$ then:

$$\int q_t(m)dm = 1.$$

**Lemma 4.** When $\alpha > 0$, for all $\epsilon > 0$, for all $\Pi_t$, there exists $r > 1$ such that when $Q_t$'s are constructed from $r$ as shown in Eq. 6, $D_\alpha(\Pi_t, Q_t) < \epsilon$

**Lemma 5.** The expected frequentist regret of the policy that constructs $Q_t$'s as in Eq. 6 and sample from $Q_t$ for $T' = \Theta(T)$ time-steps is linear and the lower bound of the average regret per time-step is

$$L = \begin{cases} c\Delta(1 - (1 - \epsilon\alpha(1-\alpha))^{\frac{1}{1-\alpha}}), & \text{when } \alpha > 1 \text{ and } 0 < \epsilon \\ c\Delta(1 - \frac{1}{e^\epsilon}), & \text{when } \alpha = 1 \text{ and } 0 < \epsilon \\ c\Delta(1 - (1 - \epsilon\alpha(1-\alpha))^{\frac{1}{1-\alpha}}), & \text{when } 0 < \alpha < 1 \text{ and } 0 < \epsilon \leq \frac{1}{\alpha(1-\alpha)} \end{cases},$$

where $c = \frac{T'}{T}$ is $\Theta(1)$.

## A.1 Proof of Lemma 3

*Proof.* First we will show that $\Pi_{t,2} = 1 - \Pi_{t,1} > 0$ for all $t \geq 0$, so that $q_t(m)$ is well-defined. We have $\Pi_{0,2} = \mathbb{P}(M_2 > M_1) > 0$ by assumption. Let $S_t = \{m \in \Omega_2 : \pi_t(m) > 0\}$ be the support of $\Pi_t$ in $\Omega_2$. If $\pi_0(m) > 0$, then $\pi_t(m) > 0$ because $\pi_t(m)$ is the product of $\pi_0(m)$ and non-zero likelihoods. Therefore $S_0 \subseteq S_t$.

Since $\mathbb{P}(M_2 > M_1) = \int_{S_0} \pi_0(m)dm > 0$, $\int_{S_0} dm > 0$. Since $S_0 \subseteq S_t$, $\int_{S_t} dm > 0$. Therefore $\int_{S_t} \pi_t(m)dm > 0$ since $S_t = \{m \in \Omega_2 : \pi_t(m) > 0\}$ by definition. Then $\Pi_{t,2} = \int_{\Omega_2} \pi_t(m)dm = \int_{S_t} \pi_t(m)dm > 0$.

Assume that $\int \pi_t(m)dm = 1$, we will show that $\int q_t(m)dm = 1$:

$$\int q_t(m)dm$$

$$= \int_{\Omega_1} q_t(m)dm + \int_{\Omega_2} q_t(m)dm$$

$$= \int_{\Omega_1} \frac{1}{r}\pi_t(m)dm + \int_{\Omega_2} \frac{1 - \Pi_{t,1}/r}{1 - \Pi_{t,1}}\pi_t(m)dm$$

$$= \frac{1}{r}\Pi_{t,1} + \frac{1 - \Pi_{t,1}/r}{1 - \Pi_{t,1}}\Pi_{t,2}$$

$$= \frac{1}{r}\Pi_{t,1} + \frac{1 - \Pi_{t,1}/r}{1 - \Pi_{t,1}}(1 - \Pi_{t,1})$$

$$= 1 .$$

$\square$

## A.2 Proof of Lemma 4

*Proof.* First we calculate the $\alpha$-divergence between $\Pi_t$ and $Q_t$ constructed in Eq. 6. Let $\Omega_1 \subseteq \Omega$ denote the region where $m_1 > m_2$ and $\Omega_2 \subseteq \Omega$ denote the region where $m_2 \geq m_1$.

When $\alpha > 0, \alpha \neq 1$ we have:

$$D_\alpha(\Pi_t, Q_t)$$

$$= \frac{1 - \int \left(\frac{\pi_t(m)}{q_t(m)}\right)^\alpha q_t(m)dm}{\alpha(1 - \alpha)}$$

$$= \frac{1 - \int_{\Omega_1} \left(\frac{\pi_t(m)}{q_t(m)}\right)^\alpha q_t(m)dm - \int_{\Omega_2} \left(\frac{\pi_t(m)}{q_t(m)}\right)^\alpha q_t(m)dm}{\alpha(1 - \alpha)}$$

$$= \frac{1 - \int_{\Omega_1} (r)^\alpha q_t(m)dm - \int_{\Omega_2} \left(\frac{1 - \Pi_{t,1}}{1 - \Pi_{t,1}/r}\right)^\alpha q_t(m)dm}{\alpha(1 - \alpha)}$$

$$= \frac{1 - Q_t(\Omega_1)(r)^\alpha - Q_t(\Omega_2)\left(\frac{1 - \Pi_{t,1}}{1 - \Pi_{t,1}/r}\right)^\alpha}{\alpha(1 - \alpha)}$$

$$= \frac{1 - \frac{\Pi_{t,1}}{r}(r)^\alpha - (1 - \frac{\Pi_{t,1}}{r})\left(\frac{1 - \Pi_{t,1}}{1 - \Pi_{t,1}/r}\right)^\alpha}{\alpha(1 - \alpha)}$$

$$= \frac{1}{\alpha(1 - \alpha)}\left(1 - \Pi_{t,1}r^{-1+\alpha} - (1 - \Pi_{t,1})^\alpha(1 - \frac{\Pi_{t,1}}{r})^{1-\alpha}\right) . \tag{7}$$

When $\alpha = 1$:

$$D_\alpha(\Pi_t, Q_t)$$

$$= \int \pi_t(m) \log\left(\frac{\pi_t(m)}{q_t(m)}\right)dm$$

$$= \int_{\Omega_1} \pi_t(m) \log\frac{\pi_t(m)}{q_t(m)}dm + \int_{\Omega_2} \pi_t(m) \log\frac{\pi_t(m)}{q_t(m)}\,dm$$

$$= \int_{\Omega_1} \pi_t(m) \log(r)dm$$

$$\quad + \int_{\Omega_2} \pi_t(m) \log\frac{1 - \Pi_{t,1}}{1 - \Pi_{t,1}/r}dm$$

$$= \Pi_{t,1} \log(r) + (1 - \Pi_{t,1}) \log\frac{1 - \Pi_{t,1}}{1 - \Pi_{t,1}/r} .$$

We will now upper bound the above expression. Consider 2 cases

- $\alpha = 1$: We have

$$D_\alpha(\Pi_t, Q_t)$$

$$= \Pi_{t,1} \log(r) + (1 - \Pi_{t,1}) \log \frac{1 - \Pi_{t,1}}{1 - \Pi_{t,1}/r}$$

$$\leq \Pi_{t,1} \log(r) + (1 - \Pi_{t,1}) \log(r) \text{ because } r > 1$$

$$\leq \log(r) .$$

- $\alpha > 0, \alpha \neq 1$: The following inequality is true by simple calculations when $0 < \alpha < 1$ or $\alpha > 1$:

$$\frac{\left(\frac{1-\Pi_{t,1}}{1-\frac{\Pi_{t,1}}{r}}\right)^{\alpha-1}}{\alpha(\alpha - 1)} \leq \frac{r^{\alpha-1}}{\alpha(\alpha - 1)} . \tag{8}$$

Then we will have:

$$D_\alpha(\Pi_t, Q_t)$$

$$= \frac{\Pi_{t,1} r^{\alpha-1} + (1 - \Pi_{t,1}) \left(\frac{1-\Pi_{t,1}}{1-\frac{\Pi_{t,1}}{r}}\right)^{\alpha-1} - 1}{\alpha(\alpha - 1)}$$

$$\leq \frac{1}{\alpha(\alpha - 1)} \left( \Pi_{t,1} r^{\alpha-1} + (1 - \Pi_{t,1}) r^{\alpha-1} - 1 \right)$$

$$= \frac{1}{\alpha(\alpha - 1)} \left( r^{-1+\alpha} - 1 \right) .$$

Therefore $D_\alpha(\Pi_t, Q_t)$ is upper bounded by:

$$\begin{cases} \frac{1-r^{\alpha-1}}{\alpha(1-\alpha)}, & \text{if } 0 < \alpha < 1 \text{ or } \alpha > 1 \\ \log(r), & \text{if } \alpha = 1 . \end{cases} \tag{9}$$

Since $\lim_{r \to 1} \log(r) = 0$ and $\lim_{r \to 1} \frac{1-r^{-1+\alpha}}{\alpha(1-\alpha)} = 0$, for any $\epsilon > 0$, there exists $r > 1$ such that

$$D_\alpha(\Pi_t, Q_t) \leq \epsilon .$$

$\square$

### A.3 Proof of Lemma 5

*Proof.* We will now lower bound the regret as a function of $\epsilon$.

For any posterior $\Pi_t$, since the approximate algorithm sampling from $Q_t$ picks the optimal arm with probability at most $1/r$ it then picks arm 2 with probability at least $1 - 1/r$.

Since we sample from $Q_t$ for $T'$ time steps, the lower bound of the average expected regret per time step is :

$$L = \frac{T'}{T}(m_1^* - m_2^*)(1 - 1/r) = c\Delta(1 - 1/r) .$$

where $\Delta = m_1^* - m_2^*$ and $c = \frac{T'}{T}$ is $\Theta(1)$.

We calculate $\epsilon$ as a function of $r$ from Eq. 9:

$$\epsilon = \begin{cases} \frac{1-r^{-1+\alpha}}{\alpha(1-\alpha)}, & \text{if } \alpha \neq 1 \\ log(r), & \text{if } \alpha = 1 . \end{cases}$$

The functions are continous when $r > 1$. Then by direct calculations when $r \to \infty$ and $r \to 1$, the domain of $\epsilon$ is:

$$0 < \epsilon \text{ when } \alpha \geq 1 .$$

$$0 < \epsilon < \frac{1}{\alpha(1 - \alpha)} \text{ when } 0 < \alpha < 1 .$$

Then

$$r = \begin{cases} (1 - \epsilon\alpha(1-\alpha))^{\frac{1}{-1+\alpha}} & \text{when } \alpha > 1 \text{ and } 0 < \epsilon \\ e^{\epsilon} & \text{when } \alpha = 1 \text{ and } 0 < \epsilon \\ (1 - \epsilon\alpha(1-\alpha))^{\frac{1}{-1+\alpha}} & \text{when } 0 < \alpha < 1 \text{ and } 0 < \epsilon \le \frac{1}{\alpha(1-\alpha)}. \end{cases}$$

Therefore we can calculate the lower bound of the regret per time-step as:

$$L = \begin{cases} c\Delta(1 - (1 - \epsilon\alpha(1-\alpha))^{\frac{1}{1-\alpha}}), & \text{when } \alpha > 1 \text{ and } 0 < \epsilon \\ c\Delta(1 - \frac{1}{e^{\epsilon}}), & \text{when } \alpha = 1 \text{ and } 0 < \epsilon \\ c\Delta(1 - (1 - \epsilon\alpha(1-\alpha))^{\frac{1}{1-\alpha}}), & \text{when } 0 < \alpha < 1 \text{ and } 0 < \epsilon \le \frac{1}{\alpha(1-\alpha)}. \end{cases}$$

We plot the lower bound of the average regret per time step when $\Delta = 0.1$ as a function of $\epsilon$ in Fig 4.

$\square$

### A.4 The Average Regret per Time-step

To understand how the constant average regret per time-step depends on $\epsilon$ and $\alpha$, we plot in Figure 4 the lower bound of the average regret per time-step in Lemma 5 as a function of $\epsilon$ in the following setting of the example constructed in the proof of Theorem 1. The algorithm samples from $Q_t$ at $T/2$ time-steps and $\Delta = 0.1$. In this case the average regret per time step is upper bounded by $\Delta/2 = 0.05$. The formula and proof are detailed in Lemma 5 in Appendix A. When $\alpha$ is around 1,

Figure 4: Lower bound of regret per time-step as a function of $\epsilon$ when $m_1^* - m_2^* = 0.1$ and we sample from the approximation for $T/2$ time-steps in the example constructed in the proof of Theorem 1. When $\alpha$ is around 1, the lower bound converges quickly as $\epsilon$ increases.

the lower bound, and therefore the average regret per time-step, converges the fastest to $\Delta/2$ as $\epsilon$ increases. When $\alpha$ is very large or close to 0, the lower bound grows slowly as $\epsilon$ increases.

### A.5 Proof of Corollary 1

Since $\mathbb{P}(M_1 > M_2) > 0$, there exist constants $\Delta > 0, \gamma > 0$ such that $\mathbb{P}(M_1 - M_2 \ge \Delta) = \gamma$. The probability that the assumption $m_1^* > m_2^*$ in Theorem 1 is satisfied is at least $\gamma > 0$. Therefore the expected regret over the prior is at least $\gamma$ times the frequentist regret in Theorem 1, which is linear.

## B Proof of Theorem 2 and Corollary 2

First we will prove Theorem 2. Let $\Pi_{t,i}$ denote $\Pi_t(\Omega_i)$. We construct the pdf of $Q_t$'s as follows:

$$q_t(m) = \begin{cases} \frac{1}{\Pi_{t,2}}\pi_t(m), & \text{if } m_2 > m_1 \\ 0, & \text{otherwise.} \end{cases} \tag{10}$$

We will prove the theorem by the following steps:

- In Lemma 6 we show that $Q_t$'s are valid distributions.

- In Lemma 7 we show that $Q_t$ has linear frequentist regret, and calculate the constant average regret per time-step.

- In Lemma 8 we show that there exists a bad prior such that the $\alpha$-divergence between $Q_t$ and $\Pi_t$ can be arbitrarily small.

In Appendix B.4 we discuss the prior-dependent error threshold $\epsilon$ that will cause linear regret. In Appendix B.5 we provide the Bayesian regret proof for Corollary 2.

**Lemma 6.** $q_t(m)$ in Eq. 10 is well-defined and if $\int \pi_t(m)dm = 1$ then:

$$\int q_t(m)dm = 1.$$

**Lemma 7.** $Q_t$ constructed in Eq. 10 chooses arm 2 at all time-steps. The average frequentist regret per time-step is $\Delta = m_1^* - m_2^*$.

**Lemma 8.** Let $\alpha < 1$, $M_1 - M_2$ and $M_2$ be independent and arm 2 be chosen at all time-steps before $t$.

For any $\epsilon > 0$, there exists $0 < z \leq 1$ such that if $\Pi_{0,2} = z$ then $D_\alpha(\Pi_t, Q_t) < \epsilon$ where $Q_t$ is constructed in Eq. 10.

For any $0 < z \leq 1$, there exists $\epsilon > 0$ such that if $\Pi_{0,2} = z$ then $D_\alpha(\Pi_t, Q_t) < \epsilon$ where $Q_t$ is constructed in Eq. 10.

## B.1 Proof of Lemma 6

*Proof.* Similar to the proof of Lemma 3, we have that $\Pi_{t,2} > 0$ for all $t \geq 0$.

Assume that $\int \pi_t(m)dm = 1$, we will show that $\int q_t(m)dm = 1$:

$$\int q_t(m)dm$$
$$= \int_{\Omega_1} q_t(m)dm + \int_{\Omega_2} q_t(m)dm$$
$$= 0 + \int_{\Omega_2} \frac{1}{\Pi_{t,2}}\pi_t(m)dm$$
$$= \frac{1}{\Pi_{t,2}} \int_{\Omega_2} \pi_t(m)dm$$
$$= 1 .$$

$\square$

## B.2 Proof of Lemma 7

*Proof.* Under the approximate distribution, arm 2 is chosen with probability 1 at all times. Clearly this approximate distribution has linear regret, with $\Delta = m_1^* - m_2^*$ being the average regret per time-step. $\square$

## B.3 Proof of Lemma 8

*Proof.* Let $D = M_1 - M_2$ which is independent of $M_2$ by the assumption. Let $f$ denote the pdf. Since the algorithm always picks arm 2, $H_{t-1}$ and $M_1$ are independent given $M_2$. Therefore for all $m_1, m_2$ and $h$, $f_{M_1|M_2,H_{t-1}}(m_1|m_2, h) = f_{M_1|M_2}(m_1|m_2)$.

Since $D = M_1 - M_2$, we have $f_{D|M_2,H_{t-1}}(m_1 - m_2|m_2, h) = f_{M_1|M_2,H_{t-1}}(m_1|m_2, h)$. Therefore for all $d, m_2$ and $h$:

$$f_{D|M_2,H_{t-1}}(d|m_2, h) = f_{M_1|M_2,H_{t-1}}(m_2 + d|m_2, h) = f_{M_1|M_2}(m_2 + d|m_2) = f_{D|M_2}(d|m_2) .$$

Since $f_{D|M_2,H_{t-1}}(d|m_2,h) = f_{D|M_2}(d|m_2)$ for all $d, m_2$ and $h$, $D$ and $H_{t-1}$ are independent given $M_2$. Then

$$
\begin{aligned}
&f_{D|M_2,H_{t-1}}(d|m_2,h) \\
=&f_{D|M_2}(d|m_2) \text{ because } D \text{ and } H_{t-1} \text{ are independent given } M_2 \\
=&f_D(d) \text{ because } D \text{ and } M_2 \text{ are independent.}
\end{aligned}
$$

Now we will show that $D$ and $H_{t-1}$ are independent. For all $d$ and $h$:

$$
\begin{aligned}
&f_{D|H_{t-1}}(d|h) \\
=& \int f_{D,M_2|H_{t-1}}(d, m_2|h) dm_2 \\
=& \int f_{D|M_2,H_{t-1}}(d|m_2, h) f_{M_2|H_{t-1}}(m_2|h) dm_2 \\
=& \int f_D(d) f_{M_2|H_{t-1}}(m_2|h) dm_2 \\
=& f_D(d) \int f_{M_2|H_{t-1}}(m_2|h) dm_2 \\
=& f_D(d) \ .
\end{aligned}
$$

Since $D$ and $H_{t-1}$ are independent, at all times $t$ the posterior does not concentrate:

$$
\Pi_{t,2} = \mathbb{P}(M_1 - M_2 < 0|H_{t-1}) = \mathbb{P}(M_1 < M_2) \ .
$$

For simplicity let

$$
z := \mathbb{P}(M_1 < M_2) \ .
$$

We will show that $D(\Pi_t, Q_t)$ is small if $z$ is large enough. First we calculate the $\alpha$-divergence between $\Pi_t$ and $Q_t$ constructed in Eq 10.

When $\alpha < 1, \alpha \neq 0$:

$$
\begin{aligned}
&D_\alpha(\Pi_t, Q_t) \\
=& \frac{1 - \int \left(\frac{q_t(m)}{\pi_t(m)}\right)^{1-\alpha} \pi_t(m) dm}{\alpha(1-\alpha)} \\
=& \frac{1 - \int_{\Omega_1} \left(\frac{q_t(m)}{\pi_t(m)}\right)^{1-\alpha} \pi_t(m) dm - \int_{\Omega_2} \left(\frac{q_t(m)}{\pi_t(m)}\right)^{1-\alpha} \pi_t(m) dm}{\alpha(1-\alpha)} \\
=& \frac{1 - 0 - \int_{\Omega_2} \left(\frac{1}{\Pi_{t,2}}\right)^{1-\alpha} \pi_t(m) dm}{\alpha(1-\alpha)} \text{ since } \alpha < 1 \\
=& \frac{1 - \left(\frac{1}{\Pi_{t,2}}\right)^{1-\alpha} \int_{\Omega_2} \pi_t(m) dm}{\alpha(1-\alpha)} \\
=& \frac{1 - \left(\frac{1}{\Pi_{t,2}}\right)^{1-\alpha} \Pi_{t,2}}{\alpha(1-\alpha)} \\
=& \frac{1 - (\Pi_{t,2})^\alpha}{\alpha(1-\alpha)} \\
=& \frac{1 - z^\alpha}{\alpha(1-\alpha)} \ .
\end{aligned}
$$

When $\alpha = 0$:

$$D_\alpha(\Pi_t, Q_t)$$

$$= \int q_t(m) \log \frac{q_t(m)}{\pi_t(m)} dm$$

$$= \int_{\Omega_1} q_t(m) \log \frac{q_t(m)}{\pi_t(m)} dm$$

$$+ \int_{\Omega_2} q_t(m) \log \frac{q_t(m)}{\pi_t(m)} dm$$

$$= \int_{\Omega_1} 0 \log(0) dm + \int_{\Omega_2} q_t(m) \log \frac{1}{\Pi_{t,2}} dm$$

$$= 0 + 1 \log \frac{1}{\Pi_{t,2}} = \log \frac{1}{\Pi_{t,2}} = \log \frac{1}{z} .$$

Note that if we don't have the condition on the prior such that picking arm 2 does not help to learn which arm is the better one, $\Pi_{t,2}$ may converge to 0, making $D_\alpha(\Pi_t, Q_t)$ goes to $\infty$ when $\alpha \leq 0$. But since $\Pi_{t,2} = z$, we will now show that for any $\alpha < 1$, for any $\epsilon > 0$, there exists $z(0 < z < 1)$ such that

$$D_\alpha(\Pi_t, Q_t) < \epsilon .$$

Consider the 2 cases

- When $\alpha < 1, \alpha \neq 0$: Since

$$\lim_{z \to 1} \frac{1 - z^\alpha}{\alpha(1 - \alpha)} = 0 .$$

  Then for any $\epsilon > 0$ there exists $0 < z < 1$ such that $D_\alpha(\Pi_t, Q_t) < \epsilon$. For any $0 < z < 1$ there exists $\epsilon > 0$ such that $D_\alpha(\Pi_t, Q_t) < \epsilon$.

- When $\alpha = 0$:

$$D_\alpha(\Pi_t, Q_t) = \log \frac{1}{z} .$$

  Since $\lim_{z \to 1} \log(1/z) = 0$, for any $\epsilon > 0$ there exists $0 < z < 1$ such that $D_0(\Pi_t, Q_t) < \epsilon$. For any $z < 1$ there exists $\epsilon > 0$ such that $D_\alpha(\Pi_t, Q_t) < \epsilon$.

$\square$

## B.4 Prior-dependent Error Threshold for Linear Frequentist Regret

In the example constructed in the previous sections, the $\alpha$-divergence between $\Pi_t$ and $Q_t$ can be calculated as: $\epsilon = \begin{cases} \frac{1-z^\alpha}{\alpha(1-\alpha)}, & \text{if } 0 < \alpha < 1 \text{ or } \alpha < 0 \\ \log \frac{1}{z}, & \text{if } \alpha = 0 \end{cases}$ .

In Figure 5, we show the values of $\epsilon$ as a function of $z$ that will make the regret linear for different values of $\alpha$. We can see that for both cases when $\alpha \leq 0$ and $0 \leq \alpha < 1$, and $z$ is not too small, there is a threshold of $\epsilon$ for each value of $z$ that makes the regret linear. For each value of $z$, if the error is smaller than the threshold we hypothesize that the regret might become sub-linear. However even if that is the case, it is not possible to calculate the exact threshold for more complicated priors. Therefore in Section 5.1 we propose an algorithm that is guaranteed to have sub-linear regret for any $\epsilon$ and any $z$ when $\alpha \leq 0$.

## B.5 Proof of Corollary 2

Since $\mathbb{P}(M_1 > M_2) > 0$, there exist constants $\Delta > 0, \gamma > 0$ such that the $\mathbb{P}(M_1 - M_2 \geq \Delta) = \gamma$. The probability that the assumption $m_1^* > m_2^*$ in Theorem 2 is satisfied is at least $\gamma > 0$. Therefore the expected regret over the prior is at least $\gamma$ times the frequentist regret in Theorem 2, which is linear.

(a) $D_\alpha(\Pi_t, Q_t) = \epsilon$ as a function of $z$ when $\alpha \leq 0$. When $z$ is very small and $\alpha$ is small, $\epsilon$ needs to be very large. When $z > 0.2$, there is a threshold of $\epsilon$ which is less than 8 that can cause linear regret.

(b) $\epsilon$ as a function of $z$ when $0 \leq \alpha < 1$. There is a threshold of $\epsilon$ which is less than 8 for each value of $z$ that can cause linear regret..

Figure 5: $\epsilon$ as a function $z$ that makes the regret linear for different values of $\alpha$ for the example constructed in the proof of Theorem 2.

## C   Proof of Lemma 2

To convert between $D_\alpha(\Pi_t, Q_t)$ and $D_\alpha(\overline{\Pi_t}, \overline{Q_t})$ we first prove the following lemma:

**Lemma 9** (Jensen's Inequality). *Let $f : \mathcal{R}^2 \to \mathcal{R}$ be a convex function. Let $P : \mathcal{R}^k \to \mathcal{R}$ and $Q : \mathcal{R}^k \to \mathcal{R}$ be 2 functions. Let $S$ is a subset of $R^k$, the domain of $x$ and $|S|$ denote the volume of $S$. Then*

$$
\frac{1}{|S|} \int_S f(P(x), Q(x))dx
$$
$$
\geq f\left(\frac{1}{|S|} \int_S P(x)dx, \frac{1}{|S|} \int_S Q(x)dx\right) . \tag{11}
$$

*Proof.* The multivariate Jensen's Inequality states that if $X$ is a n-dimensional random vector and $f : \mathcal{R}^n \to \mathcal{R}$ is a convex function then

$$
\mathbb{E}(f(X)) \geq f(\mathbb{E}(X)) .
$$

To use the multivariate Jensen's Inequality we define the 2-dimensional random vector $X : S \to \mathcal{R}^2$ by $X(x) := (P(x), Q(x))$ and a probability distribution over $S$ such that for all $x \in S$: $\mathbb{P}(x) = \frac{1}{|S|}$.

Then the left-hand side of Eq. 11 becomes $\mathbb{E}(f(X))$, while the right-hand side becomes $f(\mathbb{E}(X))$, and Eq. 11 follows from the multivariate Jensen's Inequality. $\square$

Now we will prove Lemma 2.

*Proof of Lemma 2.* Since $D_\alpha(p, q)$ is convex (Cichocki & Amari, 2010), the following functions:

$$
f(p, q) = q \log \frac{q}{p},
$$
$$
f(p, q) = p \log \frac{p}{q},
$$
$$
f(p, q) = \frac{p^\alpha q^{1-\alpha}}{\alpha(\alpha - 1)}
$$

are convex, and we can apply Lemma 9:

- When $\alpha = 0$:

$$D_\alpha(\Pi_t, Q_t)$$

$$= \int q_t(m) \log \frac{q_t(m)}{\pi_t(m)} dm$$

$$= \sum_i \int_{\Omega_i} q_t(m) \log \frac{q_t(m)}{\pi_t(m)} dm$$

$$\geq \sum_i |\Omega_i| \frac{1}{|\Omega_i|} \int_{\Omega_i} q_t(m) dm \log \frac{\frac{1}{|\Omega_i|} \int_{\Omega_i} q_t(m) dm}{\frac{1}{|\Omega_i|} \int_{\Omega_i} \pi_t(m) dm} \quad \text{by applying Lemma 9}$$

$$= \sum_i Q_{t,i} \log \frac{Q_{t,i}}{\Pi_{t,i}}$$

$$= D_\alpha(\overline{\Pi_t}, \overline{Q_t}) .$$

- When $\alpha = 1$:

$$D_\alpha(\Pi_t, Q_t)$$

$$= \int \pi_t(m) \log \frac{\pi_t(m)}{q_t(m)} dm$$

$$= \sum_i \int_{\Omega_i} \pi_t(m) \log \frac{\pi_t(m)}{q_t(m)} dm$$

$$\geq \sum_i |\Omega_i| \frac{1}{|\Omega_i|} \int_{\Omega_i} \pi_t(m) dm \log \frac{\frac{1}{|\Omega_i|} \int_{\Omega_i} \pi_t(m) dm}{\frac{1}{|\Omega_i|} \int_{\Omega_i} q_t(m) dm} \quad \text{by applying Lemma 9}$$

$$= \sum_i \Pi_{t,i} \log \frac{\Pi_{t,i}}{Q_{t,i}}$$

$$= D_\alpha(\overline{\Pi_t}, \overline{Q_t}) .$$

- When $\alpha \neq 0, \alpha \neq 1$:

$$D_\alpha(\Pi_t, Q_t)$$

$$= \int \frac{\pi(x)^\alpha q(x)^{1-\alpha} - 1}{-\alpha(1 - \alpha)} dx$$

$$= \frac{-1}{\alpha(\alpha - 1)} + \sum_i \int_{\Omega_i} \frac{\pi(x)^\alpha q(x)^{1-\alpha}}{\alpha(\alpha - 1)} dx$$

$$\geq \frac{-1}{\alpha(\alpha - 1)} + \sum_i |\Omega_i| \frac{(\frac{\Pi_{t,i}}{|\Omega_i|})^\alpha (\frac{Q_{t,i}}{|\Omega_i|})^{1-\alpha}}{\alpha(\alpha - 1)} \quad \text{by applying Lemma 9}$$

$$= \frac{-1}{\alpha(\alpha - 1)} + \sum_i \frac{\Pi_{t,i}^\alpha Q_{t,i}^{1-\alpha}}{\alpha(\alpha - 1)}$$

$$= D_\alpha(\overline{\Pi_t}, \overline{Q_t}) .$$

$\square$

## D  Proof of Theorem 3

We will prove that the frequentist regret is sub-linear for any $m^*$. If the algorithm has sub-linear frequentist regret for all values $M = m^*$, the Bayesian regret (which is the expected value over $M$) will also be sub-linear.

Without loss of generalization, let arm 1 be the best arm. From Lemma 1, since $\sum_{t=1}^\infty p_t = \infty$, we have for all arms $i$, $\sum_{t=1}^\infty P(A_t = i | H_{t-1}) = \infty$ and therefore with probability 1:

$$\lim_{t \to \infty} \Pi_{t,1} = \lim_{t \to \infty} \mathbb{P}(A^* = 1 | H_{t-1}) = 1 , \tag{12}$$

which means that the posterior probability that arm 1 is the best arm converges to 1.

We will prove the theorem by proving the following steps:

- In Lemma 10 we show that if the probability that the posterior chooses the best arm tends to 1, then the probability that the approximation chooses the best arm also tends to 1
- In Lemma 11 and Lemma 12 we show that if the probability that the approximation chooses the best arm also tends to 1 almost surely, then it has sub-linear regret with probability 1. Therefore it has sub-linear regret in expectation over the history.

**Lemma 10.** *Let $\alpha \leq 0$ and arm 1 be the true best arm. Let $\Omega_i = \{m | m_i = max(m_1, ..., m_k)\}$ be the region where arm $i$ is the best arm. If the posterior probability that arm 1 is the best arm converges to 1:*

$$\lim_{t \to \infty} \Pi_{t,1} = 1$$

*and for all $t \geq 0$:*

$$D_\alpha(\Pi_t, Q_t) < \epsilon,$$

*then the sequence $\{Q_{t,1}\}_t$ where $Q_{t,1} = \int_{\Omega_1} q_t(m)dm$ converges and*

$$\lim_{t \to \infty} Q_{t,1} = 1 \ .$$

Next we show that if the approximate distribution concentrates, then the probability that it chooses the wrong arm decreases as $T$ goes to infinity.

**Lemma 11.** *If*

$$\lim_{t \to \infty} Q_{t,1} = 1$$

*then*

$$\lim_{T \to \infty} \frac{\sum_{t=1}^T (1 - Q_{t,1})}{T} = 0 \ .$$

From Lemma 10 and Lemma 11, since $\lim_{t \to \infty} \Pi_{t,1} = 1$ with probability 1, we have $\lim_{T \to \infty} \frac{\sum_{t=1}^T (1 - Q_{t,1})}{T} = 0$ with probability 1. We will now show that the expected regret is sub-linear:

**Lemma 12.** *Let $p_t = o(1)$ be such that $\sum_{t=1}^\infty p_t = \infty$. For any number of arms $k$, any prior $\Pi_0$ and any error threshold $\epsilon > 0$, the following algorithm has $o(T)$ regret: at every time-step $t$,*

- *with probability $1 - p_t$, sample from an approximate posterior $Q_t$ such that $\lim_{T \to \infty} \frac{\sum_{t=1}^T (1 - Q_{t,1})}{T} = 0$ with probability 1, and*

- *with probability $p_t$, sample an arm uniformly at random.*

### D.1  Proof of Lemma 10

*Proof.* Let $Q_{t,i} = \int_{\Omega_i} q_t(m)dm$ and $\Pi_{t,i} = \int_{\Omega_i} \pi_t(m)dm$ . Then

$$\lim_{t \to \infty} \Pi_{t,1} = 1$$

and we want to show that $\{Q_{t,1}\}_t$ converges and

$$\lim_{t \to \infty} Q_{t,1} = 1 \ .$$

Since $D_\alpha(\overline{\Pi_t}, \overline{Q_t}) < \epsilon$ and $\lim \Pi_{t,1} = 1$ we want to show that $\limsup Q_{t,1} = 1$. By contradiction, assume that:

$$\limsup Q_{t,1} = c < 1 \ .$$

Then there exists a sub-sequence of $\{Q_{t,1}\}_t$, denoting $Q_{t_1,1}, Q_{t_2,1}, ..., Q_{t_n,1}, ..$ such that

$$\lim_{n \to \infty} Q_{t_n,1} = c . \tag{13}$$

which implies

$$0 < 1 - c = \lim_{n \to \infty} \sum_{i=2}^{k} Q_{t_n,i} \leq \sum_{i=2}^{k} \limsup_{n \to \infty} Q_{t_n,i}.$$

Therefore there exists $j \in [2, k]$ such that:

$$\limsup_{n \to \infty} Q_{t_n,j} = d > 0 .$$

Then there exists a sub-sequence of $\{Q_{t_n,j}\}_n$, denoting $Q_{t_{n_1},j}, Q_{t_{n_2},j}, ..., Q_{t_{n_m},j}, ..$ such that

$$\lim_{m \to \infty} Q_{t_{n_m},j} = d .$$

We consider the 2 cases:

- When $\alpha = 0$:

$$D_\alpha(\overline{\Pi}_t, \overline{Q}_t) = \sum_{i=1}^{k} Q_{t,i} \log \frac{Q_{t,i}}{\Pi_{t,i}} .$$

    Then we have:

$$
\begin{aligned}
\epsilon &= \lim_{m \to \infty} D_\alpha(\overline{\Pi}_{t_{n_m}}, \overline{Q}_{t_{n_m}}) \\
&\geq \lim_{m \to \infty} Q_{t_{n_m},1} \log \frac{Q_{t_{n_m},1}}{\Pi_{t_{n_m},1}} + \lim_{m \to \infty} Q_{t_{n_m},j} \log \frac{Q_{t_{n_m},j}}{\Pi_{t_{n_m},j}} \\
&= c \log \frac{c}{1} + d \log \frac{d}{0} \\
&= \infty \text{ since } d > 0,
\end{aligned}
$$

    which is a contradiction. Therefore $c = 1$.

- When $\alpha < 0$:

$$D_\alpha(\overline{\Pi}_t, \overline{Q}_t) = \frac{\sum_{i=1}^{k} \Pi_{t,i}^\alpha Q_{t,i}^{1-\alpha} - 1}{\alpha(\alpha - 1)} .$$

    Then we have:

$$
\begin{aligned}
\epsilon &= \lim_{m \to \infty} D_\alpha(\overline{\Pi}_{t_{n_m}}, \overline{Q}_{t_{n_m}}) \\
&\geq \lim_{m \to \infty} \frac{\Pi_{t_{n_m},1}^\alpha Q_{t_{n_m},1}^{1-\alpha} + \Pi_{t_{n_m},j}^\alpha Q_{t_{n_m},j}^{1-\alpha} - 1}{\alpha(\alpha - 1)} \\
&= \frac{1^\alpha c^{1-\alpha} + \frac{d^{1-\alpha}}{(0)^{-\alpha}} - 1}{\alpha(\alpha - 1)} \\
&= \infty, \text{ since } d > 0 \text{ and } \alpha < 0,
\end{aligned}
$$

    which is a contradiction. Therefore $c = 1$.

Similarly we will show that:

$$\liminf Q_{t,1} = 1 .$$

By contradiction, assume that:

$$\liminf Q_{t,1} = c' < 1 .$$

Then there exists a sub-sequence of $\{Q_{t,1}\}_t$, denoting $Q_{t_1,1}, Q_{t_2,1}, ..., Q_{t_{n'},1}, ..$ such that

$$\lim_{n \to \infty} Q_{t_{n'},1} = c' .$$

Using the same argument following Eq. 13 we will have $c' = 1$. Since $\liminf Q_{t,1} = \limsup Q_{t,1} = 1$, we have that $\{Q_{t,1}\}_t$ converges and

$$\lim Q_{t,1} = 1 .$$

$\square$

## D.2  Proof for Lemma 11

For simplicity let $x_t$ denote $1 - Q_{t,1}$. We want to show that if a sequence $\{x_t\}$ satisfies $x_t \geq 0\ \forall t$ and:

$$\lim_{t \to \infty} x_t = 0,$$

then

$$\lim_{T \to \infty} S_T = 0,$$

where

$$S_T = \frac{\sum_{t=1}^{T} x_t}{T} .$$

Since $\lim_{t \to \infty} x_t = 0$ and $x_t \geq 0\ \forall t$, for any $\epsilon > 0$ there exists $T_0$ such that for all $t > T_0$:

$$x_t < \frac{\epsilon}{2} .$$

Then for all $T > T_0$:

$$
\begin{aligned}
S_T &= \frac{x_1 + ... + x_{T_0}}{T} + \frac{x_{T_0+1} + ... + x_T}{T} \\
&\leq \frac{x_1 + ... + x_{T_0}}{T} + \frac{\frac{\epsilon}{2}T}{T} \\
&\leq \frac{x_1 + ... + x_{T_0}}{T} + \frac{\epsilon}{2} .
\end{aligned}
$$

Choose $T_1$ large enough such that $\frac{x_1 + ... + x_{T_0}}{T_1} < \frac{\epsilon}{2}$. Let $T_2 = \max(T_0, T_1)$. Then for all $T > T_2$:

$$S_T = \frac{x_1 + ... + x_{T_0}}{T} + \frac{\epsilon}{2} < \frac{\epsilon}{2} + \frac{\epsilon}{2} = \epsilon .$$

Therefore for any $\epsilon > 0$, there exists $T_2$ such that for all $T > T_2$, $S_T < \epsilon$. Since $S_T \geq 0\ \forall T$, we have:

$$\lim_{T \to \infty} S_T = 0 .$$

## D.3  Proof of Lemma 12

Without loss of generalization, let arm 1 be the true best arm. Let $\Delta = m_1^* - \max(m_2^*, ..., m_k^*)$ be the gap between the highest mean $m_1^*$ and the next highest mean of the arms.

Since $p_t = o(1)$, $\sum_{t=1}^{T} p_t$ is $o(T)$. Therefore the regret from the uniform sampling steps is $o(T)$.

Since $1 - Q_{t,1}$ is the probability of choosing a sub-optimal arm by sampling from $Q_t$, the regret of sampling from $Q_t$ is upper bounded by:

$$\mathbb{E} \sum_{t=1}^{T} \Delta(1 - Q_{t,1}) .$$

Since $\lim_{T\to\infty} \frac{\sum_{t=1}^{T}(1-Q_{t,1})}{T} = 0$ with probability 1, we have

$$\lim_{T\to\infty} \frac{\sum_{t=1}^{T} \Delta(1-Q_{t,1})}{T} = 0$$

with probability 1. Therefore

$$\lim_{T\to\infty} \mathbb{E}\frac{\sum_{t=1}^{T} \Delta(1-Q_{t,1})}{T} = 0,$$

which means that the regret of sampling from $Q_t$ is sub-linear. Since both the expected regrets of the uniform sampling steps and of sampling from $Q_t$ are sub-linear, the total expected regret is sub-linear.

## E  Ensemble Sampling and Uniform Exploration

To the best of our knowledge, (Lu & Van Roy, 2017) is the only work that provides a theoretical analysis of Thompson sampling when the sampling step is approximate. Lu & Van Roy (2017) propose an approximate sampling method called Ensemble sampling where they maintain a set of $\mathcal{M}$ models to approximate the posterior, and analyze its regret for linear contextual bandits. When the model is a $k$-armed bandit, the regret bound is as follow:

**Lemma 13** (implied by (Lu & Van Roy, 2017)). *Let $\pi^{TS}$ and $\pi^{ES}$ denote the exact Thompson sampling and Ensemble sampling policies. Let $\Delta = max(m_1^*, ..., m_k^*) - \min(m_1^*, ..., m_k^*)$. For all $\epsilon > 0$, if*

$$\mathcal{M} \geq \frac{2k}{\epsilon^2} \log \frac{2kT}{\epsilon^2\delta},$$

*then*

$$\text{Regret}(T, \pi^{ES}) \leq \text{Regret}(T, \pi^{TS}) + \epsilon\Delta T + \delta\Delta T \tag{14}$$

Lu & Van Roy (2017) prove the regret bound by only using the following property of the Ensemble sampling method: at time $t$, with probability $1 - \delta$, Ensemble sampling satisfies the following constraint:

$$\text{KL}(\overline{Q}_t, \overline{\Pi}_t) < \epsilon^2, \tag{15}$$

where $\epsilon$ is a constant if $\mathcal{M}$ is $\Theta(\log(T))$. If $\epsilon$ is a constant the regret will be linear because of the term $\epsilon\Delta T$.

At time $t$, with probability $1 - \delta$, $\text{KL}(\overline{Q}_t, \overline{\Pi}_t) < \epsilon^2$. The first 2 terms in the right hand side of Eq. 14 comes from the time-steps when $\text{KL}(\overline{Q}_t, \overline{\Pi}_t) < \epsilon^2$, and the last term comes from the other case with probability $\delta$.

Theorem 3 shows that applying an uniform sampling step will make the posterior concentrate. Moreover, Lemma 10 implies that if Eq. 15 is satisfied at a subset of times $\mathcal{T}_0 \subseteq [0, 1, ..., T]$, the approximation $Q_t$ will also concentrate when $t \in \mathcal{T}_0$. Therefore the regret from the time-steps in $\mathcal{T}_0$ will be sub-linear in $\mathcal{T}_0$, which is sub-linear in $T$.

So if we want to maintain a small number of models $M = \Theta(\log(T))$ and achieve sub-linear regret, we can apply Theorem 3 as follow. First we choose $\delta$ to be small such that the last term in Eq. 14 $\delta\Delta T$ is $o(T)$. Then we apply the uniform sampling step as shown in Theorem 3, so that the first 2 terms in the right hand side of Eq. 14 become sub-linear. We can then achieve sub-linear regret with Ensemble sampling with a $\Theta(\log T)$ number of models.

## F  KL Divergence between two Gaussian Distributions

The KL divergence between two Gaussian distributions is:

$$\text{KL}(\text{Norm}(\mu_1, \Sigma_1), \text{Norm}(\mu_2, \Sigma_2))$$
$$= \frac{1}{2}(\text{trace}(\Sigma_2^{-1}\Sigma_1) - k$$
$$+ (\mu_2 - \mu_1)^T\Sigma_2^{-1}(\mu_2 - \mu_1) + \ln\frac{\det\Sigma_2}{\det\Sigma_1})$$

# G  Posterior Calculation

In our simulations, when both the prior and the reward distributions are Gaussian, we calculate the true posterior using the following closed-form solution.

Let the posterior at time $t$ be multivariate Gaussian distribution $\mathrm{Norm}(\mu_t, \Sigma_t)$ where $\mu_t$ is a $k \times 1$ vector and $\Sigma_t$ is a $k \times k$ covariance matrix. Let the reward distribution of arm $i$ be $\mathrm{Norm}(m_i^*, \sigma^2)$ where $\sigma$ is known and $m_i^*$'s are unknown.

Let $A_t \in \{0, 1\}^k$ be a 0/1 vector where $A_t(i) = 1$ if arm $i$ is chosen at time $t$, and 0 otherwise. Let $r_t \in \mathcal{R}$ be the reward of the arm chosen at time $t$.

Then the posterior at time $t + 1$ is $\mathrm{Norm}(\mu_{t+1}, \Sigma_{t+1})$ where:

$$\Sigma_{t+1} = (\Sigma_t^{-1} + A_t A_t^T / \sigma^2)^{-1}$$
$$\mu_{t+1} = \Sigma_{t+1} (\Sigma_t^{-1} \mu_t + A_t r_t / \sigma^2) \,.$$