[Reviews · NeurIPS 2019]

Reviewer 1



Thompson Sampling with Approximate Inference ============================================= This paper investigates the performance of Thompson sampling, when the sampled distribution does not match the "problem" distribution exactly. The authors clearly explain some settings where mismatched sampling distributions can cause linear regret. The authors support their analysis with some expository "toy" experiments. There are several things to like about this paper: - This paper is one of the first to provide a clear analysis of Thompson sampling in the regime of imperfect inference. [High] - This paper is very well written, and will help build intuition and understanding for this problem and field of research. [Medium] - The authors support their claims through clear statement of theorems, experiments and do provide novel insights. [High] In some places the paper could be improved: - I do think that the emprical evaluations are overly "toy"... particularly the discussion of approximating Q_t, Z_t... this type of gross systematic mis-estimation seems like it would not occur from say "Bootstrapping" or some reasonably-updating posterior estimates? - I don't think that the addition of additional noise exploration is really the "right" approach... although that is an interesting hypothesis. It seems like another solution that would "intuitively work" is to artificially expand the "prior" of the Thompson sampling procedure, but in a way that would concentrate away with data. I suppose this is something a little like the 50/t epsilon schedule though... so I don't hate that idea. - I think that there is a thread of research on approximate TS in the RL-literature that is worth mentioning, of which "ensemble sampling" is one instance = "Deep Exploration via Randomized Value Functions" https://arxiv.org/pdf/1703.07608.pdf (or earlier ICML https://arxiv.org/pdf/1402.0635.pdf). In this, the authors show that an "approximate Gaussian" posterior can still satisfy the same regret bounds even on a mismatched Dirichlet posterior... (note that H=1 is the special case of the bandit) certainly this result is even less impressive than the Frequentist worst case guarantees of Agrawal etc. With this in mind, I do think it's a stretch to say Lu+VR are the only theoretical analysis of TS with approximate inference...The main reason I point to that work is the analysis in terms of "stochastic optimism", since that appears to suggest thatt a certain type of mis-estimation is more/less bad than others and it might be nice to connect to this alpha divergence! Overall I really like the paper, it appears crisp, insightful and polished. Some of the comparisons and observations are slightly "toy", which is the only reason I won't push ratings even higher.

Reviewer 2



For complicated probabilistic models such as Bayesian neural networks, approximate inference is inevitable and theoretical understanding of the ramification when using it in reinforcement learning are unknown. Hence this is a very relevant and interesting question. One should also point out that the recent approach of garbage-in reward-out [Kveton (2019)] solves the issue in a different manner and is clearly related to the problem authors are trying to tackle. The paper stylistics can be significantly improved. It contains duplicate sentences such as in 156-157 and includes repetitive paragraphs such as 129-143 and 194-222. The major issue I have with this paper is that as far as I understand the mapping f in e.g. Theorem 1 is the approximate inference mapping. i.e. it takes the true posterior to the approximate posterior. I do not see how existence of a bad mapping can imply that no regret is never possible. An interesting results would point out that no mapping that corresponding to approximate inference scheme can lead to no regret with the example given. The statement, as given, in my opinion cannot imply the conclusion made in the statement. Also, the posterior approximate might be close in different alphas. So this should be specified in the statement in the theorems. Simulations argue in favour of the authors argument and are nice but their presentation can be vastly improved by making them more concise. I believe that this work is a nice direction but requires significant reworking and better explanation of the theoretical statements if authors believe they are correct. Upon feedback I realize the authors aim is to show a worst case example is possible and sufficiently close in one metric is not enough to guarantee a no-regret algorithm. I believe these results are interesting for the research community but I still maintain the paper needs significant polishing.

Reviewer 3



The paper is well organized, and mostly well-written (except for many spelling/grammar mistakes, which I've pointed out below together with a correction). In my opinion, fixing these would increase the quality of writing. The studied problem is original, and of significance in practice. The authors provide a clear statement of the problem formulation as well as their contributions. I quickly checked the proofs in the supplementary, and they appear correct to me. Let me mention that I did not check all the details in the proofs. Detailed comments: 1- There is one sentence that I do not agree with (see line 52 and further). It says: "On the other hand, in the Bayesian setting, we assume that the mean is a random variable M (...), distributed according to a known prior denoted \Pi_0." That is not correct. A Bayesian agent believes that the mean (vector) takes a single value. However, the Bayesian agent does not know this value, and therefore she expresses her belief about this value in terms of a probability distribution: a prior, and therefore, the mean is treated as a random variable distributed according to the prior \Pi_0. It could be fixed by replacing the sentence above with: "On the other hand, in the Bayesian setting, an agent expresses her beliefs about the mean vector in terms of a prior \Pi_0, and therefore, the mean is treated as a random variable M (...) distributed according to the prior \Pi_0." 2- Furthermore, you might want to add in line 76: minimizing the reverse KL divergence is to some readers better known under the term "information projection", you might add "(information projection)" between brackets after "reverse KL divergence", or in the footnote. 3- Below I list the grammatical and spelling error I’ve spotted as well as some minor errors in typesetting formulas, just because it would be nice if everyone who will read this paper will not be distracted by the grammatical mistakes. line 20: with theoretical -> with a theoretical line 54: regret over the prior -> regret under the prior line 82: bandit problem. -> bandit problems. line 84: time-step -> time-steps line 92: is the prior means -> is the vector of prior means line 94: co-variance -> covariance line 95: KL divergence between 2 Gaussian -> KL divergence between two Gaussian line 102: Z_t are both larger -> Z_t is in both cases larger line 137 (and elsewhere in the manuscript): the 2 statements -> the two statements line 165: never pick the -> never picks the line 214: be such that \sum_{t}^{\infty} -> \sum_{t=1}^{\infty} line 271: in details in -> in detail in line 278: the constraints -> the constraint line 308: it seems there is a capital missing in this reference line 401: 1/r then it picks -> 1/r, it then picks line 422: m2^* -> m_2^* line 437: M_2 are independent -> M_2 be independent line 440: full stop after the sentence (i.e. (...) constructed in Eq. 10. line 442: full stop after the sentence line 453: always pick arm -> always picks arm line 460: all time t -> all times t line 467: goes to \infty -> go to \infty line 472: comma after the equation in 471, and: Then for any -> for any line 476: full stop after the sentence line 483: that make the -> that makes the line 498: Proof. Multivariate -> Proof. The Multivariate line 498: vector random variable -> random vector line 500: vector random variable -> random vector line 516: \sum_{t}^{\infty} -> \sum_{t=1}^{\infty} line 583: Q_t is sub-linear -> Q_t are sub-linear line 589: model is k-armed -> model is a k-armed line 603: sub-set -> subset line 606: of model M - > of models M line 612: between 2 Gaussian -> between two Gaussian line 612: distributions are -> distributions is line 617: co-variance -> covariance line 621: full stop (period) after the sentence. There are many full stops (period) missing after sentences ending with an equation. In particular, full stops are missing before the following lines: 55, 384, 389, 390, 392, 396, 404, 405, 441, 443, 446, 461, 462, 465, 466, 470, 475, 498, 506, 509, 510, 511, 550, 553, 556, 570, 571, 572, 574, 580, 581, 594, 613, and the end of 621. -- Update (after author feedback) -- I have read all reviews and authors' reponse. I believe the paper arguably appears to be among very few works studying such a problem in TS, and the authors provide nice and interesting results, which would pave the way for further research in this direction. The paper requires some polishing. Yet, I believe that this will not pose a significant workload, and can be accomplished within a single revision. In view of all these, I maintain my score (7), yet increase my confidence to 4.

[Author Response · NeurIPS 2019]

We thank all reviewers for their detailed feedback.

# 1 Reviewer 1

- About the solution for under-exploration: We agree that uniform exploration may not be the best possible approach. However, we think it makes the important point that adding some exploration in response to approximate inference can guarantee sub-linear regret, even though this particular form of exploration might not be optimal.

- About stochastic optimism: Thank you for the reference, we agree that stochastic optimism in RL might be related. At first glance, stochastic optimism looks quite different from a bound on the error of alpha-divergence, but it would definitely be interesting to see if there is a deeper connection.

- About experiments: We agree that $Q_t$ and $Z_t$ (which serve to illustrate the intuition about over-exploration and under-exploration) might not be natural approximations. However Figure 3 shows variational inference and ensemble sampling on a 50 armed bandit instance, which is more realistic.

# 2 Reviewer 2

- About Garbage-in Reward-out paper [Kveton, 2019]: Thank you for the reference. This paper is related but does not study the same problem – they are not concerned with the case where the Thompson sampling's inference oracle is approximate. We would cite and discuss it more thoroughly in a revision.

- About presentation issues: Thank you very much for the comments. We will revise accordingly.

- About the implication of Theorem 1: Typically, approximate inference methods minimize divergences. Broadly speaking, we show that making a divergence a small constant, alone, is not enough to guarantee sub-linear regret. We do not mean to imply that low regret is *impossible* but simply that making an alpha-divergence a small constant alone is not sufficient. We will clarify this point.

- About the implications of all theorem statements:

  Broadly speaking, Theorem 1 implies that when the approximation scheme over-explores, even though the posterior may concentrate, we suffer regret because the approximation chooses the sub-optimal arm with higher probability than the posterior at every time-step due to over-exploration.

  On the other hand, Theorem 2 implies that when the approximation scheme under-explores, the posterior may not concentrate and therefore chooses the sub-optimal arm most of the times, leading the approximation to do the same. Theorem 3 strengthens Theorem 2's observation by showing that adding forced exploration will help the posterior to concentrate and choose the optimal arm most of the times, leading the under-explored approximation to do the same.

- About the correctness of the theorem statements: We believe the results are correct. It would be great if the reviewer could point out the points of concern.

# 3 Reviewer 3

- We apologize for the grammatical mistakes and thank you so much for listing them.

- Thank you also for the suggested edits about Bayesian agent in line 52 and reverse KL divergence in line 76. We will revise accordingly.

[Meta-Review · NeurIPS 2019]

This submission was diligently evaluated, and the main problem that was identified was clarity. An additional expert reviewer was asked to checked the proofs. I am suggesting acceptance, given that the authors put a serious effort into rewriting. The reviewers did a great job in identifying where the improvements are needed, please do take advantage of that. Furthermore, for the final version, please take a look at this paper: https://link.springer.com/chapter/10.1007%2F978-3-319-46379-7_22 . The techniques used there could be relevant - please discuss this in the paper too.